# Wnt proteins can direct planar cell polarity in vertebrate ectoderm

**Chih-Wen Chu, Sergei Y Sokol\***

Department of Developmental and Regenerative Biology, Icahn School of Medicine at Mount Sinai, New York, United States

**Abstract** The coordinated orientation of cells across the tissue plane, known as planar cell polarity (PCP), is manifested by the segregation of core PCP proteins to different sides of the cell. Secreted Wnt ligands are involved in many PCP-dependent processes, yet whether they act as polarity cues has been controversial. We show that in *Xenopus* early ectoderm, the Prickle3/Vangl2 complex was polarized to anterior cell edges and this polarity was disrupted by several Wnt antagonists. In midgastrula embryos, Wnt5a, Wnt11, and Wnt11b, but not Wnt3a, acted across many cell diameters to orient Prickle3/Vangl2 complexes away from their sources regardless of their positions relative to the body axis. The planar polarity of endogenous Vangl2 in the neuroectoderm was similarly redirected by an ectopic Wnt source and disrupted after depletion of Wnt11b in the presumptive posterior region of the embryo. These observations provide evidence for the instructive role of Wnt ligands in vertebrate PCP.

## Introduction

**\*For correspondence:** sergei. sokol@mssm.edu

**Competing interests:** The authors declare that no competing interests exist.

Studies in *Drosophila* revealed the segregation of core PCP proteins to opposite sides of epithelial cells (*Goodrich and Strutt, 2011*; *Peng and Axelrod, 2012*). This mutually exclusive localization has been preserved in vertebrate tissues and is thought to be essential for multiple morphogenetic processes, including gastrulation and neurulation (*Gray et al., 2011*; *Sokol, 2015*; *Tada and Heisenberg, 2012*; *Wallingford, 2012*). Polarity cues causing the segregation of PCP complexes remain largely unknown (*McNeill, 2010*; *Wang and Nathans, 2007*). Wnt proteins have been proposed as candidates for these cues due to their involvement in many PCP-dependent processes (*Gao et al., 2011*; *Mahaffey et al., 2013*; *Ossipova et al., 2015b*; *Qian et al., 2007*; *Wu et al., 2013*; *Yang and Mlodzik, 2015*). However, whether vertebrate Wnt ligands play a permissive or instructive role in PCP remains controversial. While Wnt proteins can instruct PCP in the *Drosophila* wing and orient myocytes in chick somites (*Gros et al., 2009*; *Matis et al., 2014*; *Wu et al., 2013*), Wnt11 has been argued to act permissively in convergent extension during zebrafish gastrulation (*Heisenberg et al., 2000*).

The *Xenopus* larval epidermis contains multiciliated cells (MCCs) that are coordinately aligned to generate a unidirectional fluid flow (*Konig and Hausen, 1993*; *Werner and Mitchell, 2012*). This alignment is controlled by PCP proteins during gastrulation and neurulation (*Butler and Wallingford, 2015*; *Mitchell et al., 2009*; *Yasunaga et al., 2011*). Nevertheless, it has been challenging to document core PCP protein polarization in the ectoderm before late neurula stages (*Butler and Wallingford, 2015*; *Chien et al., 2015*; *Ciruna et al., 2006*; *Ossipova et al., 2015b*). In this study, we demonstrate that ectodermal PCP visualized by exogenous Prickle3 (Pk3)/Vangl2 complex in the epidermis and endogenous Vangl2 in the neuroectoderm can be instructed by Wnt ligands during gastrulation.

## Results and discussion

To establish early PCP markers, we examined the subcellular localization of GFP-tagged Pk3, one of the core PCP proteins predominantly expressed in the epidermal ectoderm (*Ossipova et al., 2015a*). When supplied to the ectodermal tissue by RNA microinjection, GFP-Pk3 was homogeneously distributed in the cytoplasm and at the cell junctions (*Figure 1A,B*). We hypothesized that Pk3 is not polarized because another PCP component is limiting. Given that *Drosophila* Prickle physically associates with Van Gogh (*Bastock et al., 2003*; *Jenny et al., 2003*), we suspected that the limiting factor is a Van Gogh homologue. Indeed, when GFP-Pk3 was coexpressed with Vangl2, its binding partner (*Chu et al., 2016*), epidermal PCP became evident by the beginning of neurulation with GFP fluorescence enriched at the anterior cell boundary (*Figure 1C*). In early gastrula ectoderm, GFP-Pk3 was visible as multiple membrane patches (*Figure 1D–D''*) but, at later stages, formed a single Vangl2-positive crescent- or chevron-shaped domain near the anterior cell vertex, i. e. the junction of more than two cells, with a ventral bias (*Figure 1E–H*). The anterior localization of GFP-Pk3 was confirmed by the analysis of mosaically-expressing cell clones (*Figure 1E,F*). This distribution might reflect biased stabilization of PCP proteins noted in a recent study (*Chien et al., 2015*). At the doses used, the exogenous PCP complexes did not cause any visible morphological defects. These findings establish the Pk3/Vangl2 complex as a sensor that allows direct visualization of PCP in *Xenopus* epidermal ectoderm by the end of gastrulation. This anteroposterior PCP is similar to the one observed in the *Xenopus* neural plate (*Ossipova et al., 2015b*) and other vertebrate embryonic tissues (*Antic et al., 2010*; *Borovina et al., 2010*; *Ciruna et al., 2006*; *Davey et al., 2016*; *Devenport and Fuchs, 2008*; *Hashimoto et al., 2010*; *Nishimura et al., 2012*; *Roszko et al., 2015*; *Yin et al., 2008*).

To further analyze the interaction between Pk3 and Vangl2 that is essential for their polarization, we assessed which domain is responsible for Pk3 polarity. We generated a series of Pk3 deletion mutants and examined their subcellular localization in the presence of Vangl2 (*Figure 1—figure supplement 1*). Similar to full-length Pk3, the mutated proteins did not polarize in the absence of Vangl2 (data not shown). While the N terminus of Pk3 was dispensable for its polarization, the C-terminal domain was required for Pk3 membrane recruitment, in agreement with its ability to bind Vangl2 (*Chu et al., 2016*). By contrast, Pk3 C-terminus (Pk3-C) was recruited to the plasma membrane but failed to polarize, consistent with the previous study of *Drosophila* Prickle (*Jenny et al., 2003*). A Pk3 mutant lacking the LIM domains (Pk3ΔLIM) was also unable to polarize despite being associated with the cell membrane. Of note, deletion of the CAAX motif, previously implicated in *Drosophila* Prickle polarization and stability (*Lin and Gubb, 2009*; *Strutt et al., 2013*), did not interfere with Pk3 polarization. Removal of the PET domain had a partial effect (*Figure 1—figure supplement 1A,B*), contrary to the data obtained for Prickle2 (*Butler and Wallingford, 2015*). These data show that the C-terminus is both necessary and sufficient for Vangl2-dependent membrane recruitment of Pk3, which is a prerequisite for its polarization. Notably, Pk3-C overexpression inhibited the incorporation of MCCs into the superficial epidermal cell layer at tailbud stages (data not shown), confirming the involvement of Pk3 in radial cell intercalation (*Ossipova et al., 2015a*).

Having established the utility of Pk3/Vangl2 complex as a polarity sensor, we next wanted to determine a role of Wnt signaling in ectodermal PCP. Since several Wnt ligands, including Wnt3a, Wnt5a and Wnt11b, are expressed in *Xenopus* early embryos (*Hikasa and Sokol, 2013*; *Kiecker and Niehrs, 2001*; *Ku and Melton, 1993*; *Moon et al., 1993*), we monitored GFP-Pk3/Vangl2 complex polarization in embryos, in which Wnt signaling was downregulated. Expression of the extracellular domain of Fz8 (ECD8), a potent Wnt inhibitor (*Itoh and Sokol, 1999*), disrupted Pk3/Vangl2 complex polarization in 85% of injected embryos (n = 41), whereas only 40% of control embryos lacked Pk3/Vangl2 polarity (n = 29) (*Figure 2—figure supplement 1*). Since ECD8 inhibits the majority of Wnt proteins (*Itoh and Sokol, 1999*), we utilized more selective Wnt antagonists, DN-Wnt11 and the dominant negative ROR2 receptor Ror2-TM, both of which specifically interfere with Wnt5- and Wnt11-like signals (*Bai et al., 2014*; *Hikasa et al., 2002*; *Oishi et al., 2003*; *Tada and Smith, 2000*). The majority of cells expressing DN-Wnt11 and Ror2-TM lacked GFP-Pk3 polarity in 89% (n= 28) and 88% (n= 25) of injected embryos, respectively (*Figure 2A–C*). This loss of polarity was unlikely caused by Pk3 and Vangl2 degradation, judged by immunoblotting (*Figure 2D*). Together, these experiments suggest that Wnt5- and/or Wnt11-like proteins function to establish PCP in early ectoderm.

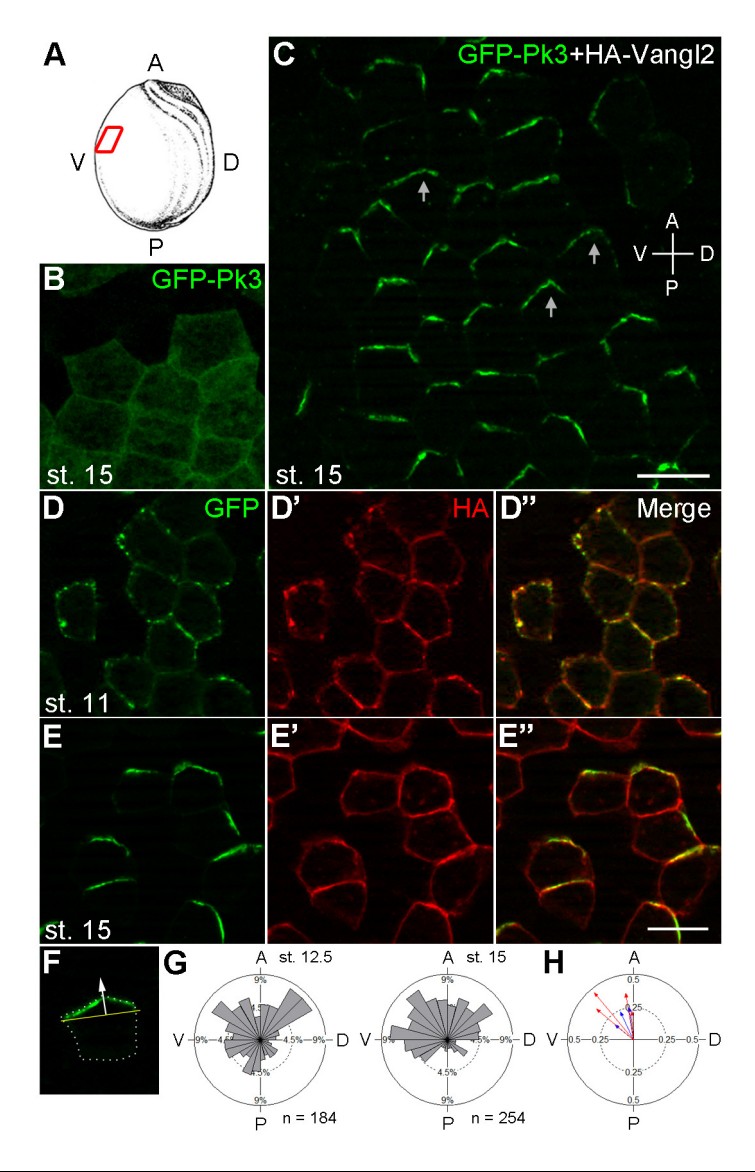

**Figure 1.** Polarization of Pk3/Vangl2 complexes at the anterior cell cortex. (**A**) Experimental scheme. RNAs encoding GFP-Pk3 and mouse HA-Vangl2 (150 pg each) were injected into the animal-ventral region of four-cell embryos. Ectoderm was dissected from the ventral midline area of fixed embryos for imaging (red box). The anteroposterior (AP) and dorsoventral (DV) axes are indicated. (**B, C**) GFP fluorescence of stage 15 ectoderm expressing GFP-Pk3 alone (**B**) or with HA-Vangl2 (**C**). Arrows in **C** point to GFP-Pk3 at the anterior cortex. (**D, E**) Embryos mosaically-expressing GFP-Pk3 and HA-Vangl2 were fixed at stage 11 (**D–D''**) or stage 15 (**E–E''**). Staining of GFP and HA is shown as indicated. HA-Vangl2 polarization is poorly detected in some cells due to variation in protein levels. Scale bar, 20 μm. (**F**) A single cell expressing GFP-Pk3 and HA-Vangl2. Dashed line depicts the cell boundary. Pk3 orientation (white arrow) is defined as perpendicular to the line connecting the ends of the crescent (yellow bar). (**G**) Rose diagrams show the orientation of GFP-Pk3 crescents in the ventral ectoderm at the indicated stage. n, number of scored cells. (**H**) Summary of GFP-Pk3 polarization derived from data in (**G**). Each arrow in the polar plot displays the mean orientation of Pk3 crescents in a single embryo at stage 12.5 (blue) or stage 15 (red). Arrow length is 1 minus the circular variance around the mean. Data are representative of two independent experiments.

The following figure supplement is available for figure 1:

**Figure supplement 1.** Different domains mediate Pk3 membrane recruitment and its polarization.

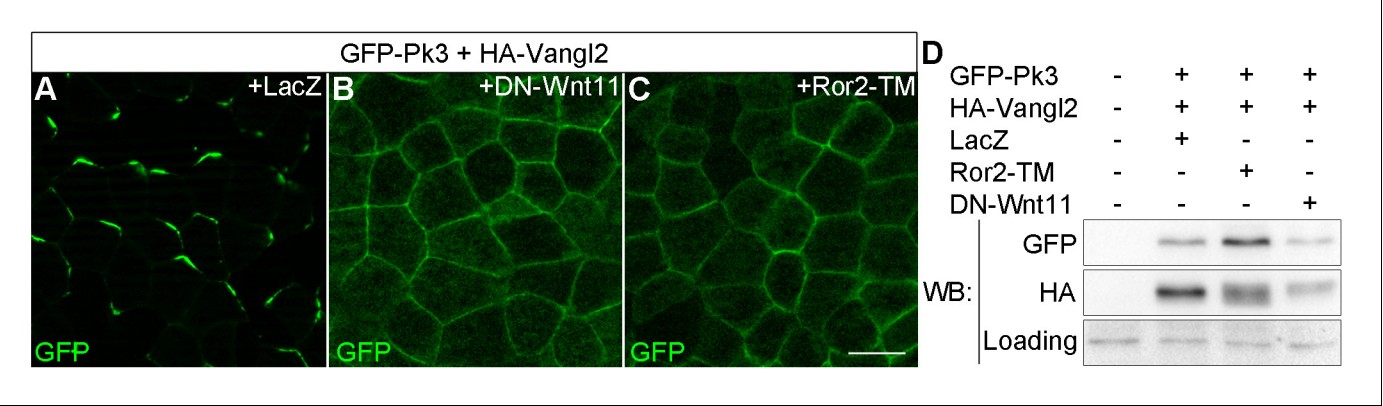

**Figure 2.** Effects of Wnt antagonists on Pk3 polarization in the epidermal ectoderm. Embryos were injected with RNAs encoding GFP-Pk3 (150 pg), *Xenopus* HA-Vangl2 (120 pg), and LacZ (1 ng, **A**) or DN-Wnt11 (2 ng, **B**) or Ror2-TM (1 ng, **C**). (**A–C**) GFP fluorescence is shown in the epidermal ectoderm of embryos fixed at stage 15. Anterior is to the top. Scale bar, 20 µm. (**D**) Protein levels of GFP-Pk3 and HA-Vangl2 in the ectoderm analyzed by immunoblotting. A non-specific band detected by anti-HA antibody reflects protein loading.

The following figure supplement is available for figure 2:

**Figure supplement 1.** ECD8 disrupts Pk3 polarization in the epidermis.

We next studied whether Wnt5a can induce ectopic Pk3 polarization in gastrula ectoderm. RNAs encoding GFP-Pk3 and Vangl2 were injected into one ventral animal blastomere of the 32-cell embryo, whereas Wnt5a RNA was coinjected with TurboFP635 (TFP) RNA as a tracer into the adjacent blastomere across the midline (*Figure 3A,B*,"L-R"). At stage 11.5, GFP-Pk3 patches formed at the cell membrane without apparent planar polarity in control embryos (*Figure 3C*). Remarkably, Wnt5a promoted early formation of polarized GFP-Pk3/Vangl2 crescents that were oriented away from the Wnt-expressing clone in 73% of injected embryos (*Figure 3D,I,J*, n = 40). These data demonstrate that Wnt5a can induce an exogenous PCP axis in non-polarized ectoderm.

To further assess whether Pk3 polarization is defined by the location of the Wnt source, we generated Wnt5a-expressing clones to the anterior of the Pk3/Vangl2-expressing clone (*Figure 3A,E*, "A-P"). By comparing the effects of Wnt5a at the lateral and anterior locations, we found that the majority of GFP-Pk3 crescents were oriented away from the Wnt5a-expressing clone regardless of its position in the ectoderm (*Figure 3B-G,I-L*). Moreover, this effect of Wnt5a persisted until neurula stages, leading to reversal of Pk3 orientation in 77% of embryos expressing Wnt5a (*Figure 3H,M,N*, n = 30). Together, these findings support the instructive role of Wnt5a in ectodermal PCP.

To find out whether the observed effect on PCP is specific to Wnt5a or can be mediated by other Wnt ligands, we evaluated the ability of Wnt3a, Wnt11 and Wnt11b, known to be expressed in the early embryo, to modulate PCP in a similar assay (*Figure 4A*). Wnt3a had little effect on GFP-Pk3 polarity (*Figure 4B,E,F*). By contrast, Wnt11 and Wnt11b behaved similarly to Wnt5a by orienting the Pk3/Vangl2 crescents away from the Wnt-expressing clone (*Figure 4C–F*). GFP-Pk3 was reoriented in 57%, 36% and 36% of the examined embryos expressing Wnt5a, Wnt11, or Wnt11b RNA, respectively (n >10). These results suggest that PCP can be instructed by these Wnt ligands, but less so by Wnt3a that acts preferentially through the β-catenin-dependent pathway (*Kikuchi et al., 2009*; *Semenov et al., 2007*).

We next attempted to find an endogenous marker or morphological structure that would provide additional evidence of early ectodermal PCP manifested by the exogenous GFP-Pk3/Vangl2 complex. Since microtubules play a critical role in PCP (*Chien et al., 2015*; *Matis et al., 2014*; *Vladar et al., 2012*), we examined the microtubule orientation at midgastrula stages by monitoring the movement of CLIP-170-GFP and EB1-GFP, two microtubule plus-end-binding proteins (*Akhmanova and Steinmetz, 2008*). In a similar experimental setting (*Figure 4A*), control embryos showed a weak alignment of CLIP170-GFP traces along the TFP clone border (*Figure 3—figure supplement 1*, *Video 1*). A slight reorientation of CLIP170-GFP traces was detected towards the border

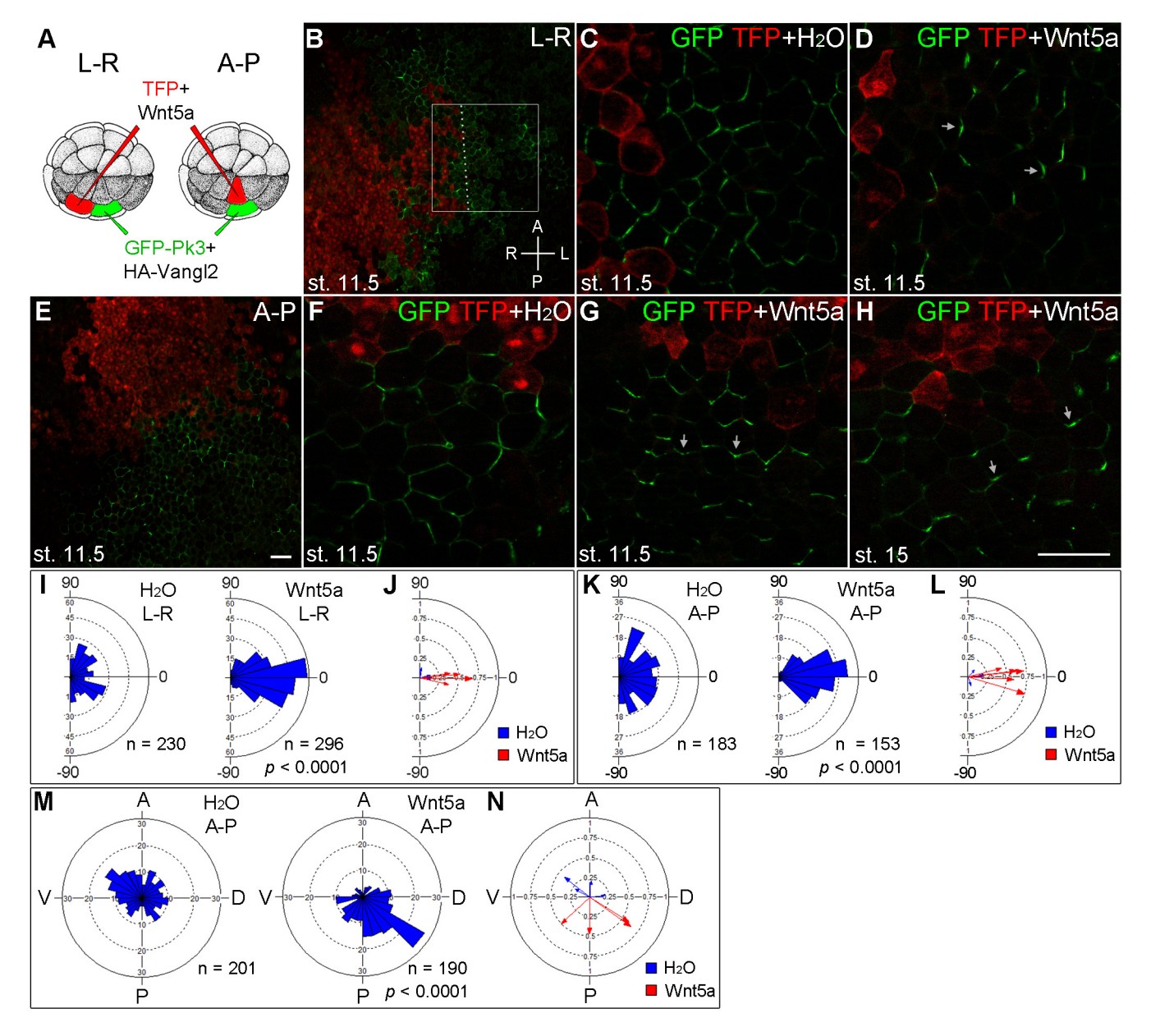

**Figure 3.** The instructive role of Wnt5a in the establishment of Pk3 polarity. (**A**) Experimental scheme. RNAs encoding GFP-Pk3 (150 pg) and *Xenopus* HA-Vangl2 (60 pg) were injected animally into a ventral blastomere of 32-cell embryos, followed by injection of TurboFP635 (TFP) RNA (150 pg, lineage tracer) with or without Wnt5a RNA (500 pg) into a blastomere either to the right (L–R) or anterior (A–P) of the Pk3-injected blastomere. The injected embryos were fixed at indicated stages, ectodermal explants were dissected, and the orientation of Pk3 crescents was evaluated by GFP fluorescence. (**B–D**) Cell orientation in L-R-positioned clones. (**B**) Low magnification view of a stage 11.5 explant. Orientation of individual cells was quantified relative to the dashed line approximating TFP clone border (boxed area). The antero-posterior and left-right axes are indicated. (**C**) Control embryo, (**D**) Wnt5a-expressing embryo. (**E–H**) Cell orientation in A–P-positioned clones at indicated stages. (**E**) Low magnification view. (**F**) Control embryo, (**G**, **H**) Wnt5a RNA-injected embryos. Arrows indicate cell orientation relative to the TFP clone (**D**, **G**, **H**). Scale bar, 50 μm. (**I**, **K**) Rose diagrams show Pk3 patch orientation in L–R (**I**) or A–P (**K**) experimental groups. Cell orientation was defined by an angle between the line joining the two ends of each Pk3 patch and the line approximating TFP clone border. (**M**) Orientation of Pk3 crescents in the A–P group at stage 15. See *Figure 1F–H* legend for quantification details. n, number of scored cells. *p* values were obtained by comparing the Wnt group to the control group using Chi-squared test. (**J**, **L**, **N**) Polar plots derived from (**I**, **K**, **M**), respectively, depict the mean Pk3 orientation in individual embryos. Arrow length is 1 minus the circular variance around the mean. Data were collected from two independent experiments.

The following figure supplements are available for figure 3:

*Figure 3 continued on next page*

*Figure 3 continued*

**Figure supplement 1.** Effect of Wnt5a on microtubule orientation.

**Figure supplement 2.** ECD8 does not direct GFP-Pk3 polarization.

of the Wnt5a clone, yet the difference was insignificant (*Figure 3—figure supplement 1*). In addition, neither live imaging of EB1-GFP nor analysis of stable microtubules visualized by the microtubule-binding protein Ensconsin-GFP revealed a significant effect of Wnt5a on microtubule alignment (data not shown). Thus, Wnt signaling might regulate core PCP proteins without reorganizing microtubules in this system, as opposed to the *Drosophila* wing (*Matis et al., 2014*). Similarly, there was no detectable bias in the position of centrosomes marked by γ-tubulin staining (data not shown). Since core PCP proteins likely represent an early response to Wnt signaling, morphological manifestations of PCP may not be fully apparent until later developmental stages.

To demonstrate the effect of Wnts on an endogenous PCP marker, we evaluated Vangl2 that is polarized in neuroectoderm but is poorly detectable in the epidermis (*Ossipova et al., 2015b*). Compared to its anterior polarization in control neuroectodermal cells, Vangl2 was reoriented away from a source of Wnt5a (*Figure 5A–D*). Such effect was observed in 90% of injected embryos (n = 46), but it was only visible in cells located one to four cell diameters away from the border of the Wnt5a clone. This finding supports our conclusions obtained for ectopic Pk3/Vangl2 complexes and suggests that the anterior polarization of Vangl2 results from endogenous Wnt proteins secreted from the posterior end of the embryo. To elucidate which Wnt ligands might be responsible, we knocked down Wnt5a and Wnt11b, two non-canonical Wnt ligands expressed at the posterior region of gastrula embryos (*Ku and Melton, 1993*; *Moon et al., 1993*), using previously characterized morpholino oligonucleotides (*Pandur et al., 2002*; *Schambony and Wedlich, 2007*). Whereas Vangl2 was accumulated at the anterior borders of cells in control embryos (87%, n = 24) and Wnt5a-depleted embryos (90%, n = 28), this polarity was retained only in 59% of embryos depleted of Wnt11b (n = 32) (*Figure 5E,F* and data not shown). These observations suggest the involvement of Wnt11b in anteroposterior PCP, consistent with its proposed activity in the gastrocoel roof plate (*Walentek et al., 2013*). Taken together, our gain- and loss-of-function assays support the idea that Wnt11b acts from the posterior region to establish an anteroposterior PCP across many cell diameters. Nevertheless, since the morpholino injection into vegetal blastomeres might partially interfere with the local production of Wnt11b in the neural plate, currently we cannot discriminate between long-range diffusion and local effects of Wnt proteins propagated by a signal relay system or cell division (*Alexandre et al., 2014*; *Farin et al., 2016*; *Zecca et al., 1996*).

Our findings support a function of Wnt5- and/or Wnt11-like proteins as biochemical polarity cues. With the demonstration that Frizzled proteins function as Wnt receptors (*Bhanot et al., 1996*), Wnt ligands were proposed to control PCP (*Adler et al., 1997*), yet no conclusion has been reached regarding the underlying mechanism (*Gros et al., 2009*; *Lawrence et al., 2002*; *Wu et al., 2013*). Wnt signaling may directly

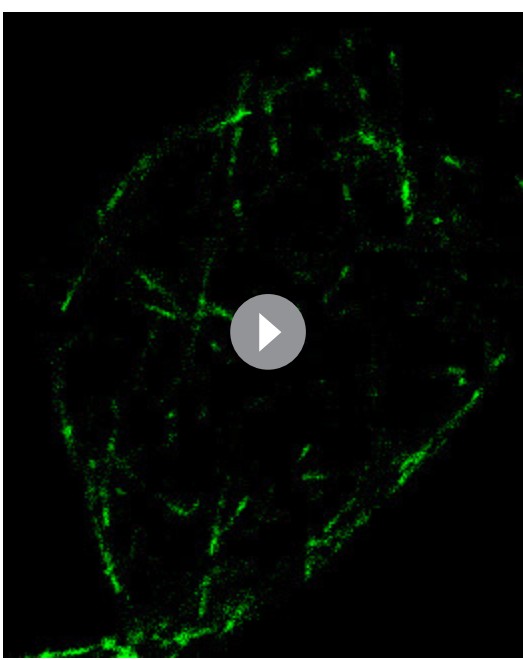

**Video 1.** Microtubule orientation visualized by the movement of Clip170-GFP foci. Time-lapse imaging of Clip170-GFP comets in an ectodermal cell of a stage 11 embryo. See *Figure 3—figure supplement 1* for details.

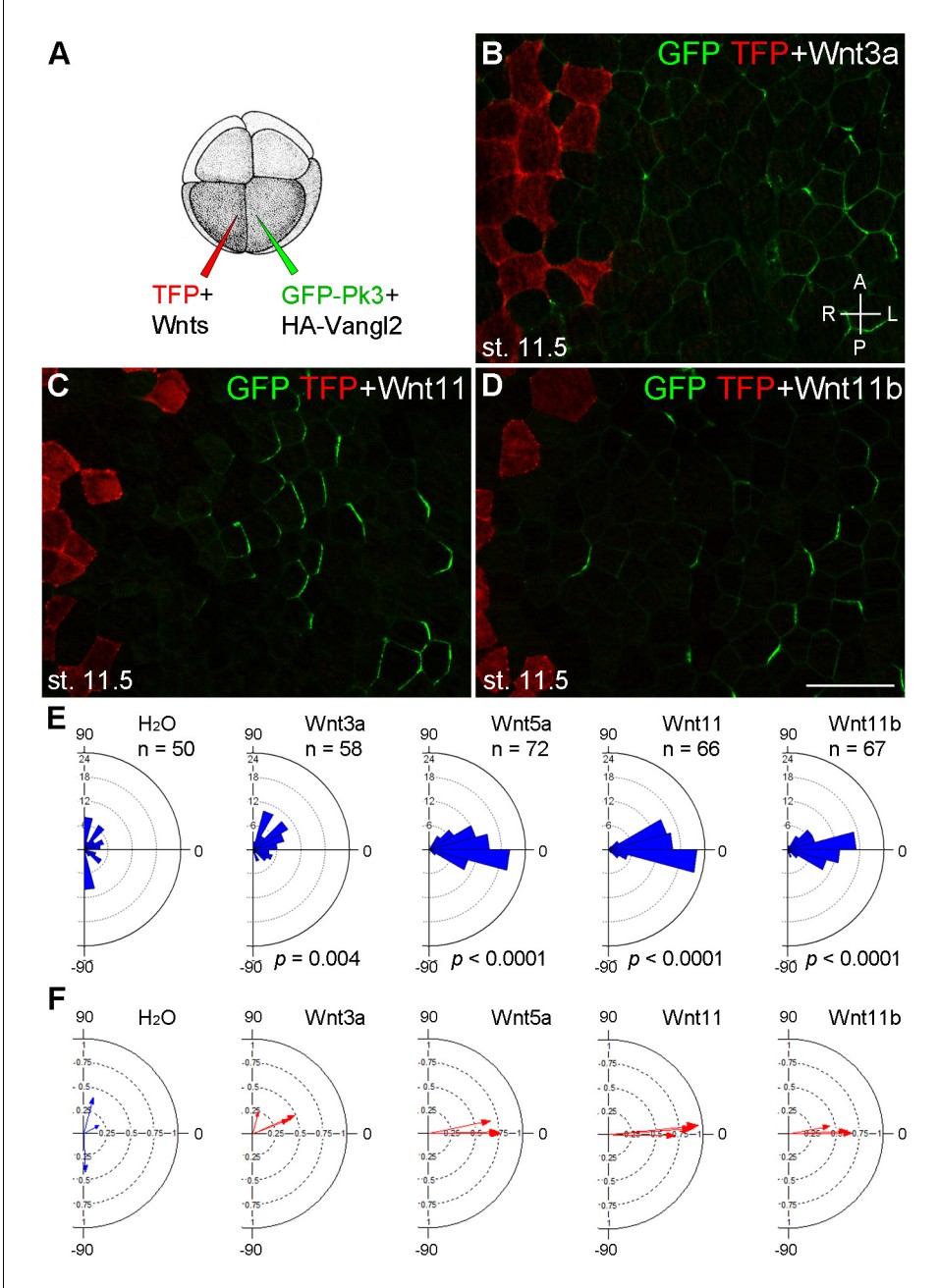

**Figure 4.** Establishment of Pk3 polarity in response to different Wnt ligands. (**A**) Experimental scheme. RNAs encoding GFP-Pk3 and mouse HA-Vangl2 (150 pg each) were injected into the left animal-ventral blastomere of eight-cell embryos, followed by coinjection of a Wnt RNA (500 pg) and TFP RNA into the right animal-ventral blastomere. (**B**–**D**) Cell orientation in stage 11.5 ectoderm of embryos injected with Wnt3a (**B**), Wnt11 (**C**) and Wnt11b (**D**) RNAs. The antero-posterior and left-right axes are indicated. Scale bar, 50 μm. (**E**) Rose diagrams show Pk3 patch orientation in clones adjacent to control ($H_2O$) or Wnt-expressing clones. See *Figure 3I* for details. n, number of scored cells. p values were obtained using Chi-squared test. (**F**) Polar plots derived from (**E**) depict the average Pk3 orientation in individual embryos. Arrow length is 1 minus the circular variance around the mean. Data were collected from two independent experiments.

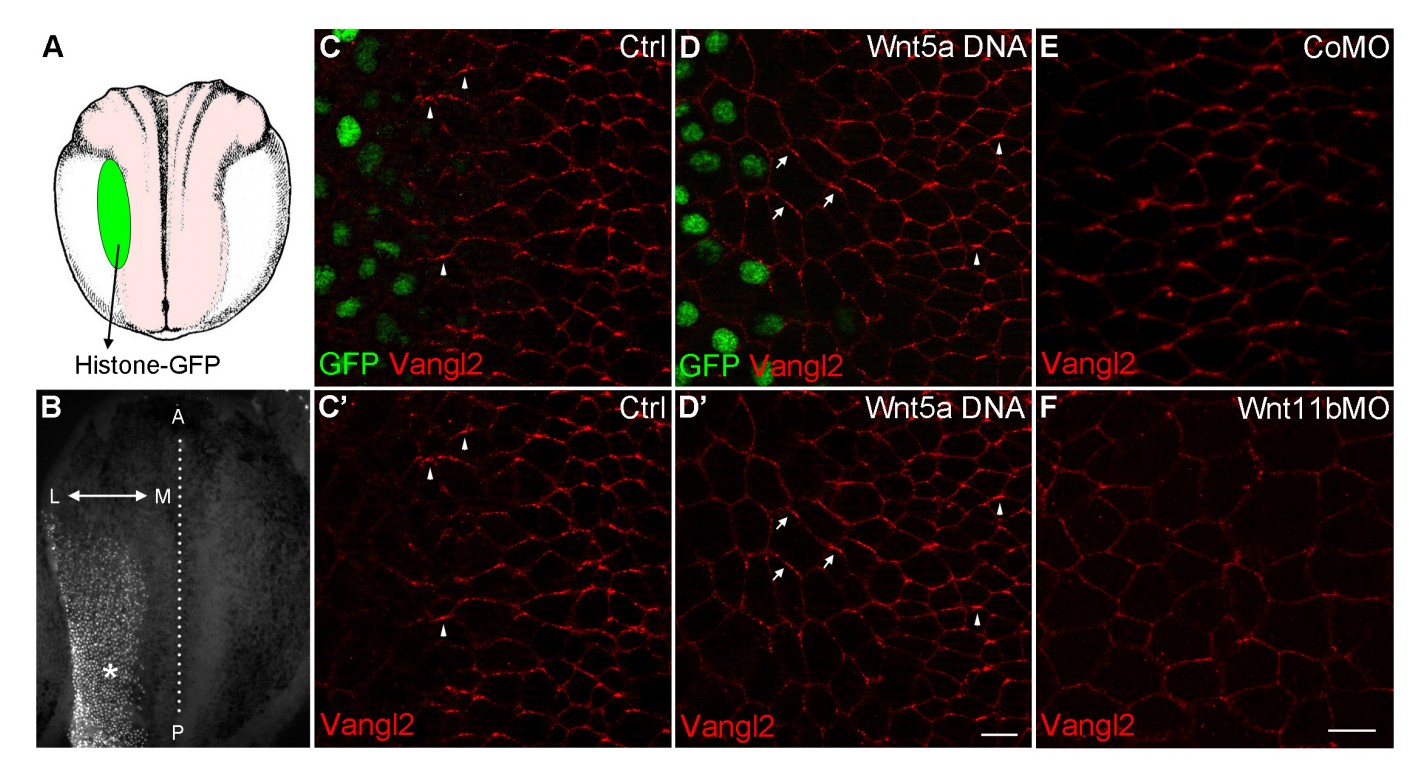

**Figure 5.** Wnt signaling instructs Vangl2 polarization in the neural plate. (**A**) Experimental scheme. Histone-GFP RNA (100 pg, nuclear lineage tracer, green) alone or with Wnt5a-Myc DNA (100 pg) was targeted to the border of the neural plate (pink), followed by immunostaining of Vangl2. (**B**) The neural plate of a stage 15 embryo with a clone of Histone-GFP-expressing cells (asterisk). Dotted line depicts the midline, and the antero-posterior (A–P) and medial-lateral (M–L) axes are indicated. (**C–D'**) Neural plates of embryos injected with Histone-GFP RNA alone (**C–C'**) or with Wnt5a-Myc DNA (**D–D'**) were immunostained for GFP and Vangl2. Vangl2 polarization is evident at the anterior cell borders (arrowheads). In cells adjacent to the Wnt5a-Myc clone, Vangl2 is oriented away from the clone (arrows). Images are representative of three independent experiments. (**E–F**) Wnt11b is required for Vangl2 polarization. Each vegetal blastomere of eight-cell embryos was injected with 20 ng of either control MO (CoMO) (**E**) or Wnt11b MO (**F**). Neural plate explants at stage 15 were stained to visualize Vangl2. Scale bar, 20 μm. Images are representative of three independent experiments.

affect core PCP proteins by regulating PCP protein post-translational modifications (*Gao et al., 2011*) or stability (*Chien et al., 2015*; *Strutt et al., 2011*). Wnt ligands were also proposed to function in PCP by blocking the Frizzled-Van Gogh interaction (*Wu and Mlodzik, 2008*). The latter explanation is less likely, because ECD8, expected to compete with Frizzled for Vangl2 binding, interfered with Pk3 polarization, instead of instructing it similar to Wnt5a (*Figure 2—figure supplement 1*, *Figure 3—figure supplement 2*). Alternatively, given the role of Wnt signaling in gastrulation (*Habas et al., 2001*), Wnt proteins might generate mechanical strains to modulate PCP (*Aigouy et al., 2010*; *Chien et al., 2015*; *Heisenberg and Bellaiche, 2013*). While the effect of mechanical forces on PCP is thought to require microtubule reorganization (*Chien et al., 2015*), we did not detect a significant change of microtubule orientation in response to Wnt5a. Although our results demonstrate that Wnt proteins can instruct Pk3 polarization in our specific experimental conditions, the immediate morphological manifestations of this activity remain obscure and whether such function involves mechanical or chemical signaling should be established by future studies.

Our observations provide support to the instructive role of Wnt proteins in PCP. By contrast, ubiquitously expressed Wnt11 can partially rescue zebrafish embryos with a mutation in the *wnt11* gene (*Heisenberg et al., 2000*). Whereas this finding suggests a permissive effect, lack of complete rescue may be also explained by the absence of proper instructions. At present, it is still unclear whether the proposed instructive mechanism operates to direct PCP during normal embryonic development.

## Materials and methods

### Plasmids, mRNA synthesis and morpholinos

GFP-Pk3, GFP-Pk3-C and GFP-Pk3ΔPET in pXT7 have been described (*Chu et al., 2016*; *Ossipova et al., 2015a*). All Pk3 deletion mutants were obtained by PCR and subcloned into pXT7-GFP. The following Pk3 constructs were made: ΔN (69–538), ΔC (1–372), C (373–538), ΔPET (deletion of amino acids 69–170), ΔLIM (deletion of 179–372), ΔCAAX is missing the last 4 amino acids. Numbers in parentheses refer to amino acid position deduced from the cDNA clone (GenBank accession number BC154995). HA-tagged *Xenopus* Vangl2 in pCS2 was generated by PCR. Details of cloning are available upon request. Wnt5a-myc was subcloned into pCS2 from a plasmid obtained from R. Moon (unpublished).

Capped mRNAs were synthesized using mMessage mMachine kit (Ambion, Austin, TX) from the linearized DNA templates encoding Pk3 derivatives and the following previously described plasmids: mouse HA-Vangl2 (*Gao et al., 2011*), Wnt3a (*Wolda et al., 1993*), Wnt5a (*Moon et al., 1993*), Wnt11/Wnt11R (*Garriock et al., 2005*)(a gift of P. Krieg), Wnt11b (*Tada and Smith, 2000*), extracellular domain of Frizzled8 (ECD8) (*Itoh and Sokol, 1999*), Ror2-TM (*Hikasa et al., 2002*), DN-Wnt11 (*Tada and Smith, 2000*). Human histone H2B-GFP-pCS2 was a gift of P. Skourides and Chenbei Chang. TurboFP635-pCS2 was made from the TurboFP635 (Katushka) plasmid obtained from A. Zaraisky.

The following morpholinos (MOs) were used: standard control oligo (CoMO) (Gene Tools), Wnt5a MO (*Schambony and Wedlich, 2007*) and Wnt11b MO (*Pandur et al., 2002*).

### *Xenopus* embryo culture and microinjections

In vitro fertilization and culture of *Xenopus laevis* embryos were carried out as previously described (*Dollar et al., 2005*). Staging was according to (*Nieuwkoop and Faber, 1994*). For microinjections, embryos were transferred into 3% Ficoll in 0.5 × MMR buffer and 5–10 nl of mRNA mixture or morpholinos was injected into one or more blastomeres. Amounts of injected mRNA per embryo have been optimized in preliminary dose-response experiments (data not shown) and are indicated in Figure legends.

### Immunoblot analysis

For protein analysis, five stage 15 embryos expressing Pk3 deletion mutants were lysed in the buffer containing 50 mM Tris-HCl pH 7.6, 50 mM NaCl, 1 mM EDTA, 1% Triton X-100, 10 mM NaF, 1 mM $Na_3VO_4$, 25 mM β-glycerol phosphate, 1 mM PMSF. For analysis of Pk3 and Vangl2, animal caps were dissected at stage 10 and incubated in 0.6 x MMR until the equivalent of stage 15 when they were lysed. After centrifugation at 15,000 g, the supernatant was subjected to SDS-PAGE and Western blot analysis using standard techniques as described (*Itoh et al., 1998*). Sample loading was controlled by staining with Ponceau S (Sigma, St. Louis, MO). Chemiluminescence was captured by the ChemiDoc MP imager (BioRad, Hercules, CA).

### Immunofluorescence, image analysis and quantification

For GFP and TFP fluorescence and immunofluorescence staining, embryos were manually devitellinized, ectoderm was dissected and fixed with MEMFA (0.1 M MOPS, pH 7.4, 2 mM EGTA, 1 mM $MgSO_4$ and 3.7% formaldehyde) for 30 min at room temperature. Indirect immunofluorescence staining was performed as described previously (*Ossipova et al., 2014*). The following primary antibodies were used: rabbit anti-Vangl2 (1:100, (*Ossipova et al., 2015b*)), mouse anti-GFP (B-2, 1:200, Santa Cruz Biotechnology, Dallas, TX) and rabbit anti-HA (1:3000, Bethyl Labs. Montgomery, TX). Secondary antibodies were Alexa Fluor 488-conjugated (1:400, Invitrogen, Waltham, MA) or Cy3-conjugated (1:400, Jackson ImmunoResearch). Stained explants were mounted for observation in the Vectashield mounting medium (Vector Labs, Burlingame, CA). Images were captured using a Zeiss AxioImager microscope with the Apotome attachment (Zeiss, Germany). Images shown are representative of 2–4 independent experiments with 6–8 embryos per group.

To quantify cell orientation, we selected embryos with clearly separable Wnt- and Pk3-expressing clones with the expected position relative to the body axis. At stage 15, scoring was done only for the cells displaying unambiguous GFP-Pk3 signal as a single crescent. Cell orientation was defined

by an arrow perpendicular to the line connecting the ends of each Pk3 crescent and quantified by ImageJ (NIH). Since Pk3 forms patches rather than crescents in stage 11.5 embryos, cell polarity was quantified differently. In this case, cell orientation was defined as an angle between the line approximating each Pk3 patch and the line tangential to TFP clone border and was measured by ImageJ. Data were collected from GFP-Pk3-expressing cells within 10 cell diameters from the TFP clone border. Rose diagrams were drawn using Oriana 3 (Kovach Computing Services, UK), and two-sample Chi-squared test was used for statistical analysis. The mean vector of Pk3 orientation per embryo was presented by polar plots.

## Microtubule end-tracking

Microtubule polarity was visualized in embryos injected with Clip170-GFP RNA, synthesized from the pCS2-CLIP170-GFP plasmid (*Werner et al., 2011*). The movement of Clip170-GFP comets was assessed under a Zeiss LSM 880 confocal microscope with a 63X oil objective. Videos of individual cells were taken at a rate of 1 frame/2.5 s and contained 12 frames. The data were processed using ImageJ. Each video was temporally color-coded to define microtubule polarity, and the angle of Clip170-GFP traces relative to the TFP clone border was measured. Oriana 3 was used to plot rose diagrams and calculate mean axial vectors of individual embryos from mean axial vectors of individual cells.

## Acknowledgement

We thank Kyeongmi Kim for the HA-tagged *Xenopus* Vangl2 construct, Brian Mitchell, Chris Kintner, Randy Moon, Chenbei Chang, Paris Skourides, and Andrey Zaraisky, for plasmids. We also thank Chi Pak for comments on the manuscript, Vladimir Gelfand for advice on microtubule plus end tracking and members of the Sokol laboratory for discussions. Confocal microscopy was performed at the Microscopy CORE at the Icahn School of Medicine at Mount Sinai. This study was supported by NIH grants to SS.

## Additional information

### Funding

| Funder | Grant reference number | Author |
| --- | --- | --- |
| National Institutes of Health | HD078874 | Sergei Y Sokol |

The funders had no role in study design, data collection and interpretation, or the decision to submit the work for publication.

### Author contributions

C-WC, Conception and design, Acquisition of data, Analysis and interpretation of data, Drafting or revising the article; SYS, Conception and design, Analysis and interpretation of data, Drafting or revising the article

### Author ORCIDs

Sergei Y Sokol, http://orcid.org/0000-0002-3963-9202

### Ethics

Animal experimentation: This study was carried out in strict accordance with the recommendations in the Guide for the Care and Use of Laboratory Animals of the National Institutes of Health. The protocol 04-1295 was approved by the institutional animal care and use committee (IACUC) of the Icahn School of Medicine at Mount Sinai.

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
