## [Decision Letter]

Thank you for submitting your article "Wnt proteins direct planar cell polarity axis in the vertebrate epidermis" for consideration by *eLife*. Your article has been reviewed by three peer reviewers, and the evaluation has been overseen by a Reviewing Editor and Janet Rossant as the Senior Editor. One of the three reviewers has agreed to reveal her identity: Cecilia Moens (Reviewer #1).

The reviewers have discussed the reviews with one another and the Reviewing Editor has drafted this decision to help you prepare a revised submission.

As stated above, the manuscript has been reviewed by three expert reviewers, and their assessments, together with my own (Jeremy Nathans, Reviewing Editor), forms the basis of this letter. I am including the three reviews at the end of this letter, as there are various specific comments in them that will not be repeated in the summary here.

All of the reviewers were impressed with the importance of your work. Overall, the experiments appear to be carefully executed. However, there is a general consensus that the data should be more cautiously interpreted and that various experiments that could strengthen (or weaken) the conclusions are feasible.

In particular, the experiments are really a demonstration of what is possible with an experimental perturbation, but not necessarily what actually happens in vivo, which would require a true loss-of-function experiment. Describing it as such is important. Also, the connection between polarized PCP protein distribution and some biological structure (e.g. cytoskeleton or cilia) is not demonstrated here. Finally, there are technical caveats related to over-expressing a Frizzled CRD or dominant Wnt that should be critically addressed.

We would like to encourage you to resubmit a revised manuscript that addresses the specific issues raised in the reviews and summarized above. We realize that several of the suggestions may be beyond the scope of the present work (e.g. doing a true loss of function analysis), but we have not removed them from the reviews since they convey the reaction to the work over-all.

*Reviewer #1:*

The planar cell polarity pathway (PCP) polarizes cells in the plane of an epithelium. Core components of the pathway include Fzd and Dsh family members, however the requirement for a Wnt ligand in planar polarization is variable: essential for some PCP processes and apparently dispensable for others. Even in contexts where a Wnt is essential, the loss of function experiments that demonstrate this do not distinguish between a permissive versus an instructive role in specifying the direction of polarization. This brief manuscript demonstrates that the polarization of PCP components occurs in response to a Wnt ligand. Using the robust Vangl2-dependent localization of Pk3 to anterior cell boundaries in the *Xenopus* non-neural ectoderm as a direct readout of PCP signaling in dominant-negative and gain-of-function experiments, they show a) that planar polarization depends on Wnt signaling and b) that localized expression of a Wnt ligand can repolarize epithelial cells several cell diameters away from the source. This is not the only mechanism by which PCP can be established; in other contexts, PCP can arise in response to tension, flow, or by unknown but Wnt-independent means. Nevertheless, the paper shows clearly that a directional Wnt signal can establish PCP and as such is an important addition to the literature. With the relatively minor revisions suggested below I feel that the paper is appropriate for publication in *eLife*.

Criticisms:

1) The quantification and statistics were done on a per-cell basis, with each N being the orientation of Pk3 in a single cell. However, it is not reasonable to expect that the behavior of one cell in an embryo is independent of the behavior of a neighboring cell in the same embryo. This is particularly true when we are dealing with planar polarity. Therefore, the data should be re-analyzed on a per-embryo basis, with each N being the average Pk3 polarization angle in a single embryo.

2) In the Results and Discussion the paper says: "Moreover, this effect of Wnt5a persisted to the neurula stage, resulting in PCP reversal in 77% of embryos”. How was this eventual polarity reversal demonstrated? Was it based on something other than the orientation of Pk crescents? This claim of tissue polarity reversal should be substantiated using a separate readout such as primary cilium orientation.

*Reviewer #1 (Additional data files and statistical comments):*

No additional data required. Statistics should be re-done on a per-embryo (rather than per-cell) basis.

*Reviewer #2:*

The central contention of this manuscript is that Wnt's provide a long range directional signal for PCP in the gastrula stage *Xenopus* ectoderm. The authors use coexpression of Pk3-GFP and Vangl2 as markers of GFP, and dominant negative Wnt and FzECD constructs to indicate a requirement for Wnt's, and Wnt ectopic expression to demonstrate the ability to determine directionality. The notion that Wnt's act as long range directional signals for PCP has been unsettled in all systems in which it has been addressed for as long as the idea has been floated. The evidence presented here that this is the case in the gastrula stage ectoderm is as good as any in any system so far, which is to say intriguing, but not entirely compelling. It has been clear for some time that Wnts *can* direct polarity. The key question is whether they *do* direct polarity, and it is here that this study, like several previous ones, falls just short of being entirely convincing. I'm on the fence about whether the arguments presented here warrant publication in a leading journal, but at a minimum there are several things that should be done to shore up the argument.

The interaction between Pk3 and Vangl2 is consistent with our understanding of PCP signaling and is not particularly novel, and the deletion analysis to identify required Pk3 domains is a bit of a distraction. Why Vangl2 should be limiting for Pk3 localization at these stages is an interesting and more puzzling question in light of recently published results. The failure to observe asymmetry of exogenous Pk3-GFP if endogenous Pk3 is asymmetrically localized is best explained if the Pk3-GFP is present in large excess, so that asymmetry that would otherwise be evident is swamped out by randomly localized and unassociated Pk3. The less interesting possibility is that endogenous Pk3 is not localized asymmetrically, but can be induced to do so in the presence of the necessary other players.

This contrasts with results from Butler and Wallingford, who showed that Pk2, Vangl1 and Dvl1 all localize asymmetrically when independently expressed at later stages. The puzzle here is the contrasting dynamics. Butler's results suggest a much later acquisition of polarity, as their probes were symmetric as late as stage 19, and only subsequently showed increasing asymmetry by the time of MCC differentiation. From these results, we would infer that there is little or no activity of the PCP components at the gastrula stage.

If that is the case, this raises the question of whether the authors are examining what is essentially the potential to polarize, or in other words, a synthetic polarization that can be created by expressing needed components (and that can then perhaps respond to endogenous or exogenous Wnt's), but that doesn't reflect an endogenous polarization that normally occurs. For example, we have no evidence that Pk3 is needed for this polarity, and there is no independent readout of polarization. In the GRP, Pk3 is normally at the basal bodies, though it can be induced to apicolateral asymmetry, and its role and localization in radial intercalation of MCCs (Sokol lab) are both potentially consistent with different mechanisms. Given these observations, it is plausible that the authors have created a synthetic PCP system that can respond to Wnt's, but that doesn't normally exist. The authors offer no other readout for PCP at this time.

On the other hand, asymmetry, or at least asymmetric activity, of Vangl2, Celsr1, Fzd3 was inferred by FRAP at this time by Chien et al. (although these assays also required the co-expression of exogenous components), and this was associated with polarized apical microtubules. However, these authors concluded that tissue strain provided the directionality to polarization, in contrast to the current contention that Wnt signaling directs polarization. If Chien et al. are correct, it may be that microtubules determine the direction of the PCP orientation, as opposed to being responsive to PCP protein polarization. Furthermore, these authors showed that microtubule orientation is responsive to strain, which might be providing the directional signal. If the authors suggest that microtubules are reading Wnt dependent polarity, it would stretch the imagination to conceive that strain reorganizes the pattern of Wnt expression. A more harmonious conclusion would be that if Wnt's are involved, they are involved in local cell to cell signaling rather than long range signaling.

Therefore, whether PCP signaling normally occurs, what if anything it is doing, and the upstream directional signal are not entirely established. Furthermore, Pk3 may or may not be involved, though even if not, it could be a legitimate marker of PCP.

The relevance of the Pk3/Vangl2 complex as a PCP marker in reading a Wnt signal would be bolstered if the authors could show that the polarized microtubules observed at this time depend on Pk3 or Vangl2. Even better, the authors should look at microtubules in their loss and gain of function experiments to determine their potential responsiveness to these inputs. If microtubules reorganize in response to Wnt's, they should then demonstrate that the requirement for normal microtubule orientation, proposed to be Wnt dependent, is lost upon knockdown of PCP components. If microtubules do not depend on Wnt endogenous or exogenous Wnt signals, then I would not feel confident that endogenous PCP signaling is occurring at gastrula stages.

Based on the currently provided results, the authors' statement "In this study, we use the complex of Pk3 and Vangl2 as a sensor of epidermal PCP to demonstrate that the polarity of this complex is directly defined by Wnt ligands during gastrulation" should instead say: "the polarity of this complex CAN BE directly defined by…".

Another important concern is the conclusion that can or cannot be drawn from the methods used to interfere with endogenous Wnt signaling. As discussed above, these are critical to the distinction between CAN and DO. The mechanisms by which the Fz8ECD and the dominant negative Wnt's act is not certain. While they may bind and block Wnt activity, it is also hypothesized that PCP involves direct interaction between Fz and Vangl proteins, and these reagents could potentially interfere with such an interaction to disrupt PCP. The authors could potentially address this concern if they could show that RNAi of the suspect Wnt's, either alone or simultaneously, would disrupt asymmetry of their PCP markers.

Last, the authors acknowledge that they don't truly distinguish between a long range Wnt signal providing a directional signal, and a paracrine signal involved in cell to cell signaling, as proposed by Adler many years ago, that would not provide a directional signal. The localized Wnt expression patterns (particularly since they are very low resolution old in situ studies) are not enough to force the former conclusion. Does this not undermine their thesis?

As is, I'm left to conclude that PCP signaling might be occurring at gastrula stage, this polarization might do something that matters, and Wnt's might provide a long range signal for this polarization. It would be good to feel a little more confident.

*Reviewer #2 (Additional data files and statistical comments):*

No additional data files needed, and assuming representative images are shown, the results as shown are compelling.

*Reviewer #3:*

The results presented support a role for Wnts in early PCP establishment in the *Xenopus* epidermis. The authors contend that this is an instructive role, and such a finding would stand in contrast to the recent report by Chien et al. (2015 Curr Biol 25, 2774-2784) that suggests mechanical cues are responsible for early PCP orientation in the *Xenopus* epidermis (although the authors don't make this point). However, as it stands, I think that while the authors make a case that Wnts *can* re-orient polarity, they do not provide good evidence that this normally happens and/or whether it happens when Wnts are expressed at physiological levels. It is equally likely that the observed results could be a dominant negative effect caused by Wnt binding to a Fzd CRD domain and blocking interactions with Vangl, similar to what Wu et al. 2013 see in *Drosophila* upon Wnt over expression.

The authors need better evidence to convince me that they have made any advance upon the previous view that Wnts might only play a permissive role in this context.

Other major comments:

1) This is not the first evidence for instructive role of Wnts in vertebrates, as Wnt5a has previously been shown to have an instructive role in the mouse limb bud, in a study that also impressively showed direct evidence of a Wnt-induced activity gradient (Gao et al. (2011) Dev Cell 20, 163-176).

2) Although the "fluorescent sensor for planar polarity in *Xenopus*" may be new (as indicated in the "Impact statement"), it is not really novel for a GFP fusion to be used to reveal polarity, as even in *Xenopus* this has been previously demonstrated (Butler and Wallingford (2015) Development 142, 3429-3439).

3) A structure-function dissection of Pk3 domains required for polarization is described, without any reference to the similar results for Pk2 described in Butler et al. 2015. I'm also concerned about how much can be concluded if endogenous protein is not first removed in a structure-function experiment like this.

[Editors' note: further revisions were requested prior to acceptance, as described below.]

Thank you for submitting your revised manuscript "Wnt proteins can direct planar cell polarity in vertebrate ectoderm" to *eLife*. The revised manuscript has been reviewed by two expert reviewers, and their assessments, together with my own (Jeremy Nathans, Reviewing Editor), forms the basis of this letter. I am including the two reviews at the end of this letter, as the specific suggestions in them will not be repeated in the summary here.

We would like to encourage you to make some additional text modifications to the manuscript that addresses the specific issues raised in the two reviews below – especially the issues raised by reviewer #2. With respect to the writing, we appreciate that in the current publishing environment there is an almost irresistible temptation to over-state one's conclusions. *eLife* is trying to restore some balance to the world of scientific writing. We welcome self-critical comments and we believe that such comments actually enhance the reader's ability to judge the science fairly.

I will offer two examples of scientific writing that I think illustrates the preceding point. They are from two papers on ubiquitin that form the core of the discoveries for which the authors shared the Nobel Prize.

Proposed role of ATP in protein breakdown: conjugation of protein with multiple chains of the polypeptide of ATP-dependent proteolysis.

Hershko A, Ciechanover A, Heller H, Haas AL, Rose IA.

Proc Natl Acad Sci U S A. 1980 Apr;77(4):1783-6.

In the second paragraph of the Discussion section, after summarizing the evidence that leads the authors to propose the conjugation of APF-1 (=ubiquitin) to proteins as a way of marking them for degradation, the authors write: "Evidence that APF-1-proteins are intermediates in the breakdown of denatured protein as proposed in Figure 6 is indirect and inconclusive at this time."

Activation of the heat-stable polypeptide of the ATP-dependent proteolytic system.

Ciechanover A, Heller H, Katz-Etzion R, Hershko A.

Proc Natl Acad Sci U S A. 1981 Feb;78(2):761-5.

In the last paragraph of the Discussion section, after describing their discovery and characterization of ubiquitin (=APF-1) ligase, the authors write "Evidence suggesting the role of the APF-1-activating enzyme in conjugation and in ATP-dependent protein breakdown is not conclusive at present."

*Reviewer #1:*

The authors have added new experiments that address some of the criticisms but they have not been able to address others. In response to the criticism that their gain-of-function experiments show that Wnt signaling CAN direct planar polarity but not that it DOES direct endogenous polarity, they include new data showing that the polarization of endogenous Vangl2 is altered by localized WNT overexpression, and that morpholino knockdown of Wnt11b eliminates endogenous Vangl2 polarization. This, combined with the gain-of-function data showing that Wnt expressing cells can repolarize Vangl2 and Pk3 over several cell diameters, is compelling evidence for a role for Wnts in orienting PCP in the frog gastrula non-neural ectoderm. On the other hand, the authors tried but failed to detect polarization of any sub-cellular structure apart from PCP core components themselves. They argue that microtubule or centrosomal asymmetry may be a much later consequence of PCP, not detectable at these early stages. They have also addressed some relatively minor comments I had about quantification of the data. In sum, the paper is improved by demonstrating a requirement for Wnts in endogenous Vangl2 localization, but remains focused on the polarity proteins themselves and not their morphological or cellular consequences. As I was already quite positive about the paper, which I found to be the clearest demonstration to date of a role for Wnts in directing epithelial PCP, I feel that it is now appropriate for publication in *eLife*.

*Reviewer #2:*

This resubmission contains all of the same data regarding the epidermal ectoderm from the first submission, unchanged except for the additional statistical analysis, plus one new figure about polarity in the neuroectoderm. As such, my opinion about the interpretive difficulties concerning the epidermal ectoderm remains unchanged. The main response of the authors was to change what they choose to say about the epidermal ectoderm data. For this part of the manuscript, the message has been changed (mostly) from the claim that a Wnt signal is instructive for planar polarization, to the well justified statement that a Wnt signal can be instructive for planar polarization.

The new data concern a separate and possibly distinct planar polarization event, so it is not clear how this is supposed to support the findings in the first four figures as opposed to representing a different example. This new data, which was presumably added to address concerns about a true loss of function phenotype, demonstrates that Wnt11b is required for planar polarity of Vangl2 in the neuroectoderm. On the other hand, it does not show that this signal works at a distance, since descendants of the 8 ventral blastomeres into which the MO was injected contribute to the neural plate, and expression of Wnt11b, perhaps at a low level, throughout the neural plate is not ruled out. On the other hand, they do show that ectopic Wnt5a can non-autonomously repolarize neural plate over several cell diameters. They acknowledge that this might occur by either paracrine signaling and propagation, or by graded signaling of the ligand. Concerning this new data, the authors are a bit more circumspect, but shockingly, the final conclusion of the paper again incorrectly claims that Wnt's *are* instructive for PCP: "Whereas our results demonstrate the instructive role of Wnt proteins in PCP…"

If we conflate the three different paradigms (ectodermal ectoderm, Wnt11b in neural ectoderm and Wnt5a in neural ectoderm), the data would probably make the best argument yet that a Wnt signal acts instructively to determine PCP, the main weaknesses being that the localized expression of Wnt needs to be more carefully examined, and one would like to see gain and loss of function for the same Wnt (or combination of Wnts) in the same tissue. However, I don't feel it's sufficiently rigorous to combine these systems to come to a conclusion as important as the claim that a Wnt IS instructive for PCP.

If the authors wish to try to produce a more definitive conclusion, I'd advise them to build a careful analysis of Wnt11b in the neural plate, and not just add the minimal data presented here as an addendum. If they wish to moderate their overall conclusion to Wnts *can* instruct planar polarity (two misleading statements would need to be changed, see below), that would be well supported by the data provided. This course would require a somewhat detailed discussion of the distinction between these conclusions. On the other hand, that would be a far less novel result, and one could debate whether it would be of sufficient impact for *eLife*.

The two unjustified statements:

"Based on these experiments, Wnt5a appears to act as an instructive cue for Pk3 polarization."

This implies that this is what it normally does.

"Whereas our results demonstrate the instructive role of Wnt proteins in PCP, whether such function involves mechanical or chemical signaling remains to be clarified in future studies."

This is clearly not supported by the present data.

It's also relevant to comment on one of the responses that the authors provided in the rebuttal.

"Several lines of evidence argue for Wnt proteins acting on ectodermal PCP over a long distance during vertebrate gastrulation. First, the effect of ectopic Wnt proteins on the GFP-Pk3/Vangl2 complex is detectable across many cell diameters."

So too is clonal knockdown of fz in a fly wing detected several cell diameters away. But this would still clearly be a paracrine signal that is then propagated via other means beyond the clone border. This is therefore not strong evidence for a direct long range action of a ligand.

"Second, Wnt5a can reorient endogenous Vangl2, which is normally polarized throughout the neuroectoderm along the anteroposterior axis (new Figure 5)."

Doesn't address the paracrine vs. long range issue.

"Third, Wnt11b depletion in the posterior region of the embryo disrupts Vangl2 polarity in the neural plate (new Figure 5), supporting its role in long-range anteroposterior PCP."

This may be the strongest argument, but still the experimental design is flawed (as noted above), as the vegetal blastomeres of eight cell embryos contribute to neural plate, so the knockdown is presumably distributed throughout the region, and local expression is not ruled out. The authors therefore correctly state "Nevertheless, currently we cannot discriminate between long-range diffusion and local effects of Wnt proteins propagated by a signal relay system or cell division (Alexandre et al., 2014; Farin et al., 2016; Zecca et al., 1996)."

Finally, if the authors wish to build a case around Wnt11b. How do they reconcile their story with "Wnt11 has been argued to act permissively in convergent extension during zebrafish gastrulation (Heisenberg et al., 2000)"? Of course, then one is comparing fish and frogs.

---

## [Author Response]

*All of the reviewers were impressed with the importance of your work. Overall, the experiments appear to be carefully executed. However, there is a general consensus that the data should be more cautiously interpreted and that various experiments that could strengthen (or weaken) the conclusions are feasible.*

*In particular, the experiments are really a demonstration of what is possible with an experimental perturbation, but not necessarily what actually happens in vivo, which would require a true loss-of-function experiment. Describing it as such is important. Also, the connection between polarized PCP protein distribution and some biological structure (e.g. cytoskeleton or cilia) is not demonstrated here. Finally, there are technical caveats related to over-expressing a Frizzled CRD or dominant Wnt that should be critically addressed.*

*We would like to encourage you to resubmit a revised manuscript that addresses the specific issues raised in the reviews and summarized above. We realize that several of the suggestions may be beyond the scope of the present work (e.g. doing a true loss of function analysis), but we have not removed them from the reviews since they convey the reaction to the work over-all.*

*Reviewer #1:*

*The planar cell polarity pathway (PCP) polarizes cells in the plane of an epithelium. Core components of the pathway include Fzd and Dsh family members, however the requirement for a Wnt ligand in planar polarization is variable: essential for some PCP processes and apparently dispensable for others. Even in contexts where a Wnt is essential, the loss of function experiments that demonstrate this do not distinguish between a permissive versus an instructive role in specifying the direction of polarization. This brief manuscript demonstrates that the polarization of PCP components occurs in response to a Wnt ligand. Using the robust Vangl2-dependent localization of Pk3 to anterior cell boundaries in the Xenopus non-neural ectoderm as a direct readout of PCP signaling in dominant-negative and gain-of-function experiments, they show a) that planar polarization depends on Wnt signaling and b) that localized expression of a Wnt ligand can repolarize epithelial cells several cell diameters away from the source. This is not the only mechanism by which PCP can be established; in other contexts, PCP can arise in response to tension, flow, or by unknown but Wnt-independent means. Nevertheless, the paper shows clearly that a directional Wnt signal can establish PCP and as such is an important addition to the literature. With the relatively minor revisions suggested below I feel that the paper is appropriate for publication in eLife.*

*Criticisms:*

*1) The quantification and statistics were done on a per-cell basis, with each N being the orientation of Pk3 in a single cell. However, it is not reasonable to expect that the behavior of one cell in an embryo is independent of the behavior of a neighboring cell in the same embryo. This is particularly true when we are dealing with planar polarity. Therefore, the data should be re-analyzed on a per-embryo basis, with each N being the average Pk3 polarization angle in a single embryo.*

As requested by the referee, we modified Figure 1, Figure 3 and Figure 4 to present the results as polar plots summarizing mean cell orientation per embryo. In addition, we kept the original rose plots to preserve raw data.

*2) In the Results and Discussion the paper says: "Moreover, this effect of Wnt5a persisted to the neurula stage, resulting in PCP reversal in 77% of embryos”. How was this eventual polarity reversal demonstrated? Was it based on something other than the orientation of Pk crescents? This claim of tissue polarity reversal should be substantiated using a separate readout such as primary cilium orientation.*

In order not to over-interpret our results, we revised the text to provide more precise description of the experiment (Results and Discussion, fifth paragraph). Prompted by the referee, we searched for other markers that indicate tissue polarization in our system and examined microtubules and centrosome polarity. Confocal live imaging of CLIP170-GFP, a microtubule plus-end-binding protein, allowed us to analyze microtubule orientation in response to Wnt5a, but we did not see a significant effect (new Figure 3—figure supplement 1, Video 1). The analysis of EB1-GFP, Ensconsin-GFP and γ-tubulin immunostaining also failed to reveal a detectable effect of Wnt5a (data not shown). Whereas Wnt signaling may trigger an early response in core PCP proteins, tissue polarity may not become visible morphologically until later developmental stages. Nevertheless, we were successful in demonstrating the Wnt5a-mediated reorientation of Vangl2 that is highly enriched in the neuroectoderm. We show that endogenous Vangl2 accumulates at the distal ends of the cells that respond to Wnt5a (new Figure 5), supporting our conclusions obtained for exogenous Pk3 and Vangl2 constructs.

*Reviewer #1 (Additional data files and statistical comments):*

*No additional data required. Statistics should be re-done on a per-embryo (rather than per-cell) basis.*

As stated above, both original image analysis and new data are now shown in modified Figure 1, Figure 3, Figure 4, following the referee’s request.

*Reviewer #2:*

*The central contention of this manuscript is that Wnt's provide a long range directional signal for PCP in the gastrula stage Xenopus ectoderm. The authors use coexpression of Pk3-GFP and Vangl2 as markers of GFP, and dominant negative Wnt and FzECD constructs to indicate a requirement for Wnt's, and Wnt ectopic expression to demonstrate the ability to determine directionality. The notion that Wnt's act as long range directional signals for PCP has been unsettled in all systems in which it has been addressed for as long as the idea has been floated. The evidence presented here that this is the case in the gastrula stage ectoderm is as good as any in any system so far, which is to say intriguing, but not entirely compelling. It has been clear for some time that Wnts can direct polarity. The key question is whether they do direct polarity, and it is here that this study, like several previous ones, falls just short of being entirely convincing. I'm on the fence about whether the arguments presented here warrant publication in a leading journal, but at a minimum there are several things that should be done to shore up the argument.*

*The interaction between Pk3 and Vangl2 is consistent with our understanding of PCP signaling and is not particularly novel, and the deletion analysis to identify required Pk3 domains is a bit of a distraction. Why Vangl2 should be limiting for Pk3 localization at these stages is an interesting and more puzzling question in light of recently published results. The failure to observe asymmetry of exogenous Pk3-GFP if endogenous Pk3 is asymmetrically localized is best explained if the Pk3-GFP is present in large excess, so that asymmetry that would otherwise be evident is swamped out by randomly localized and unassociated Pk3. The less interesting possibility is that endogenous Pk3 is not localized asymmetrically, but can be induced to do so in the presence of the necessary other players.*

*This contrasts with results from Butler and Wallingford, who showed that Pk2, Vangl1 and Dvl1 all localize asymmetrically when independently expressed at later stages. The puzzle here is the contrasting dynamics. Butler's results suggest a much later acquisition of polarity, as their probes were symmetric as late as stage 19, and only subsequently showed increasing asymmetry by the time of MCC differentiation. From these results, we would infer that there is little or no activity of the PCP components at the gastrula stage.*

*If that is the case, this raises the question of whether the authors are examining what is essentially the potential to polarize, or in other words, a synthetic polarization that can be created by expressing needed components (and that can then perhaps respond to endogenous or exogenous Wnt's), but that doesn't reflect an endogenous polarization that normally occurs. For example, we have no evidence that Pk3 is needed for this polarity, and there is no independent readout of polarization. In the GRP, Pk3 is normally at the basal bodies, though it can be induced to apicolateral asymmetry, and its role and localization in radial intercalation of MCCs (Sokol lab) are both potentially consistent with different mechanisms. Given these observations, it is plausible that the authors have created a synthetic PCP system that can respond to Wnt's, but that doesn't normally exist. The authors offer no other readout for PCP at this time.*

As the referee acknowledges in the next paragraph, our data are consistent with the observations of Chien et al., 2015, arguing for the existence of PCP at late gastrula stages. At present, we have no explanation for the discrepancy between our findings and those of Butler and Wallingford (2015). This could be due to the functional difference between PCP proteins or the difference in developmental stages. We argue that Pk3 is a relevant PCP protein, based on its expression analysis and its requirement for Vangl2 polarity and ciliogenesis (Chu *et al.*, 2016). Moreover, our new experiments that assess endogenous Vangl2 in the neuroectoderm provide further support to our claim that the distribution of GFP-Pk3 reflects endogenous PCP (new Figure 5).

*On the other hand, asymmetry, or at least asymmetric activity, of Vangl2, Celsr1, Fzd3 was inferred by FRAP at this time by Chien et al. (although these assays also required the co-expression of exogenous components), and this was associated with polarized apical microtubules. However, these authors concluded that tissue strain provided the directionality to polarization, in contrast to the current contention that Wnt signaling directs polarization. If Chien et al. are correct, it may be that microtubules determine the direction of the PCP orientation, as opposed to being responsive to PCP protein polarization. Furthermore, these authors showed that microtubule orientation is responsive to strain, which might be providing the directional signal. If the authors suggest that microtubules are reading Wnt dependent polarity, it would stretch the imagination to conceive that strain reorganizes the pattern of Wnt expression. A more harmonious conclusion would be that if Wnt's are involved, they are involved in local cell to cell signaling rather than long range signaling.*

*Therefore, whether PCP signaling normally occurs, what if anything it is doing, and the upstream directional signal are not entirely established. Furthermore, Pk3 may or may not be involved, though even if not, it could be a legitimate marker of PCP.*

*The relevance of the Pk3/Vangl2 complex as a PCP marker in reading a Wnt signal would be bolstered if the authors could show that the polarized microtubules observed at this time depend on Pk3 or Vangl2. Even better, the authors should look at microtubules in their loss and gain of function experiments to determine their potential responsiveness to these inputs. If microtubules reorganize in response to Wnt's, they should then demonstrate that the requirement for normal microtubule orientation, proposed to be Wnt dependent, is lost upon knockdown of PCP components. If microtubules do not depend on Wnt endogenous or exogenous Wnt signals, then I would not feel confident that endogenous PCP signaling is occurring at gastrula stages.*

The results of Chien et al. do not exclude a role for Wnt signaling in the establishment of PCP. Consistent with their report, either Wnt11b or other Wnts that are present at the vegetal/posterior pole may be responsible for the tension produced by the blastopore ring during gastrulation. Alternatively, one can envisage that both mechanical forces and Wnt signaling provide input to PCP.Our additional analysis of microtubule orientation in gastrula ectoderm cells does not support the idea that Wnt5a orients Pk3/Vangl2 complexes by regulating microtubule alignment (Figure 3—figure supplement 1). More detailed studies are required to elucidate the relationship between Wnt signaling and mechanical strains during gastrulation, which is beyond the scope of the current work.

*Based on the currently provided results, the authors' statement "In this study, we use the complex of Pk3 and Vangl2 as a sensor of epidermal PCP to demonstrate that the polarity of this complex is directly defined by Wnt ligands during gastrulation" should instead say: "the polarity of this complex CAN BE directly defined by…".*

We amended our statement as suggested by the referee.

*Another important concern is the conclusion that can or cannot be drawn from the methods used to interfere with endogenous Wnt signaling. As discussed above, these are critical to the distinction between CAN and DO. The mechanisms by which the Fz8ECD and the dominant negative Wnt's act is not certain. While they may bind and block Wnt activity, it is also hypothesized that PCP involves direct interaction between Fz and Vangl proteins, and these reagents could potentially interfere with such an interaction to disrupt PCP. The authors could potentially address this concern if they could show that RNAi of the suspect Wnt's, either alone or simultaneously, would disrupt asymmetry of their PCP markers.*

We used multiple Wnt antagonists including Ror2-TM, a construct interfering with Ror2 function, and tried to avoid overstating our conclusions. Prompted by the referee, we knocked down candidate *wnt5a* and *wnt11b* gene functions with previously characterized morpholino oligonucleotides. Depletion of Wnt11b disrupted Vangl2 polarity, supporting its role in PCP (Figure 5). Despite this promising observation, we feel that further analysis of the requirement of Wnt ligands in PCP is beyond the scope of our manuscript.

*Last, the authors acknowledge that they don't truly distinguish between a long range Wnt signal providing a directional signal, and a paracrine signal involved in cell to cell signaling, as proposed by Adler many years ago, that would not provide a directional signal. The localized Wnt expression patterns (particularly since they are very low resolution old in situ studies) are not enough to force the former conclusion. Does this not undermine their thesis?*

*As is, I'm left to conclude that PCP signaling might be occurring at gastrula stage, this polarization might do something that matters, and Wnt's might provide a long range signal for this polarization. It would be good to feel a little more confident.*

Several lines of evidence argue for Wnt proteins acting on ectodermal PCP over a long distance during vertebrate gastrulation. First, the effect of ectopic Wnt proteins on the GFP-Pk3/Vangl2 complex is detectable across many cell diameters. Second, Wnt5a can reorient endogenous Vangl2, which is normally polarized throughout the neuroectoderm along the anteroposterior axis (new Figure 5). Third, Wnt11b depletion in the posterior region of the embryo disrupts Vangl2 polarity in the neural plate (new Figure 5), supporting its role in long-range anteroposterior PCP. However, given the history of the attempts to resolve this issue in the *Drosophila* model in the past 20 years, we admit that further studies are required to determine whether Wnt proteins act locally or by diffusing over long distance in our system.

*Reviewer #3:*

*The results presented support a role for Wnts in early PCP establishment in the Xenopus epidermis. The authors contend that this is an instructive role, and such a finding would stand in contrast to the recent report by Chien et al. (2015 Curr Biol 25, 2774-2784) that suggests mechanical cues are responsible for early PCP orientation in the Xenopus epidermis (although the authors don't make this point).*

The results of Chien et al. do not exclude a role for Wnt signaling in the establishment of PCP as we also stated in our response to referee 2. Either Wnt11b or other Wnts that are present at the vegetal/posterior pole may be required for mechanical strains produced during gastrulation. Alternatively, both mechanical forces and Wnt signaling may regulate PCP by different mechanisms.

*However, as it stands, I think that while the authors make a case that Wnts can re-orient polarity, they do not provide good evidence that this normally happens and/or whether it happens when Wnts are expressed at physiological levels. It is equally likely that the observed results could be a dominant negative effect caused by Wnt binding to a Fzd CRD domain and blocking interactions with Vangl, similar to what Wu et al. 2013 see in Drosophila upon Wnt over expression.*

*The authors need better evidence to convince me that they have made any advance upon the previous view that Wnts might only play a permissive role in this context.*

Our new loss-of-function experiments demonstrate the involvement of Wnt11b in establishing anteroposterior PCP (Figure 5), supporting the idea that Wnt-dependent PCP happens normally in vivo. Moreover, the activity of ectopic Wnt5a in our experiments is comparable to that of the endogenous cue that leads to Vangl2 polarity in the neuroectoderm (Figure 5), but the physiological levels of active Wnt proteins are hard to assess. Although the referee suggested that Wnt5a acts in our experiments by blocking Frizzled-Vangl2 interaction, we consider it unlikely, because the extracellular domain of Frizzled 8 (ECD8), which is expected to compete with Frizzled for Vangl2 binding, did not orient the Pk3/Vangl2 complexes like Wnt5a did (Figure 3—figure supplement 2). This point has been added to the amended discussion. Overall, we argue for the instructive role of Wnts in PCP based on the ability of Wnt5a-expressing clones to orient the Pk3/Vangl2 complex regardless of their position relative to the body axis (Figure 3). This model is further supported by the effect of ectopic Wnt5a on endogenous Vangl2 in the neuroectoderm (Figure 5).

*Other major comments:*

*1) This is not the first evidence for instructive role of Wnts in vertebrates, as Wnt5a has previously been shown to have an instructive role in the mouse limb bud, in a study that also impressively showed direct evidence of a Wnt-induced activity gradient (Gao et al. (2011) Dev Cell 20, 163-176).*

Gao et al. (2011) visualized graded polarization of Vangl2 in the mouse limb and have demonstrated its disruption in Wnt5a -/- embryos. Although the authors propose an instructive role for Wnt5a, this hypothesis has not been supported by an in vivo gain-of-function assay, such as the one provided by our study.

*2) Although the "fluorescent sensor for planar polarity in Xenopus" may be new (as indicated in the "Impact statement"), it is not really novel for a GFP fusion to be used to reveal polarity, as even in Xenopus this has been previously demonstrated (Butler and Wallingford (2015) Development 142, 3429-3439).*

We agree that fluorescent GFP fusions are common tools for cell polarity studies. Nevertheless, we wish to emphasize that our constructs are the first to reveal PCP in early non-neural ectoderm. The sentence noted by the referee has been rephrased to clarify the meaning.

*3) A structure-function dissection of Pk3 domains required for polarization is described, without any reference to the similar results for Pk2 described in Butler et al. 2015.*

We apologize for inadvertently omitting this reference. This point has now been corrected.

I'm also concerned about how much can be concluded if endogenous protein is not first removed in a structure-function experiment like this.

Since we know little about how Pk3 is polarized, our initial observations may be useful even in the presence of endogenous Pk3. Future studies are needed to study the role of endogenous Pk3 and other Prickle family members in PCP.

[Editors' note: further revisions were requested prior to acceptance, as described below.]

*We would like to encourage you to make some additional text modifications to the manuscript that addresses the specific issues raised in the two reviews below – especially the issues raised by reviewer #2. With respect to the writing, we appreciate that in the current publishing environment there is an almost irresistible temptation to over-state one's conclusions. eLife is trying to restore some balance to the world of scientific writing. We welcome self-critical comments and we believe that such comments actually enhance the reader's ability to judge the science fairly.*

*I will offer two examples of scientific writing that I think illustrates the preceding point. They are from two papers on ubiquitin that form the core of the discoveries for which the authors shared the Nobel Prize.*

Proposed role of ATP in protein breakdown: conjugation of protein with multiple chains of the polypeptide of ATP-dependent proteolysis.

Hershko A, Ciechanover A, Heller H, Haas AL, Rose IA.

*Proc Natl Acad Sci U S A. 1980 Apr;77(4):1783-6.*

*In the second paragraph of the Discussion section, after summarizing the evidence that leads the authors to propose the conjugation of APF-1 (=ubiquitin) to proteins as a way of marking them for degradation, the authors write: "Evidence that APF-1-proteins are intermediates in the breakdown of denatured protein as proposed in Figure 6 is indirect and inconclusive at this time."*

Activation of the heat-stable polypeptide of the ATP-dependent proteolytic system.

Ciechanover A, Heller H, Katz-Etzion R, Hershko A.

*Proc Natl Acad Sci U S A. 1981 Feb;78(2):761-5.*

*In the last paragraph of the Discussion section, after describing their discovery and characterization of ubiquitin (=APF-1) ligase, the authors write "Evidence suggesting the role of the APF-1-activating enzyme in conjugation and in ATP-dependent protein breakdown is not conclusive at present."*

*Reviewer #1:*

*The authors have added new experiments that address some of the criticisms but they have not been able to address others. In response to the criticism that their gain-of-function experiments show that Wnt signaling CAN direct planar polarity but not that it DOES direct endogenous polarity, they include new data showing that the polarization of endogenous Vangl2 is altered by localized WNT overexpression, and that morpholino knockdown of Wnt11b eliminates endogenous Vangl2 polarization. This, combined with the gain-of-function data showing that Wnt expressing cells can repolarize Vangl2 and Pk3 over several cell diameters, is compelling evidence for a role for Wnts in orienting PCP in the frog gastrula non-neural ectoderm. On the other hand, the authors tried but failed to detect polarization of any sub-cellular structure apart from PCP core components themselves. They argue that microtubule or centrosomal asymmetry may be a much later consequence of PCP, not detectable at these early stages. They have also addressed some relatively minor comments I had about quantification of the data. In sum, the paper is improved by demonstrating a requirement for Wnts in endogenous Vangl2 localization, but remains focused on the polarity proteins themselves and not their morphological or cellular consequences. As I was already quite positive about the paper, which I found to be the clearest demonstration to date of a role for Wnts in directing epithelial PCP, I feel that it is now appropriate for publication in eLife.*

The seventh paragraph of the Results and Discussion described our experiments that attempted to document morphological consequences of Wnt-dependent PCP. So far, these experiments remain inconclusive. To acknowledge this deficiency, we introduced the following sentence in the last but one paragraph of the discussion. “Although our results demonstrate that Wnt proteins can instruct Pk3 polarization in our specific experimental conditions, the immediate morphological manifestations of this activity remain obscure”

*Reviewer #2:*

*This resubmission contains all of the same data regarding the epidermal ectoderm from the first submission, unchanged except for the additional statistical analysis, plus one new figure about polarity in the neuroectoderm. As such, my opinion about the interpretive difficulties concerning the epidermal ectoderm remains unchanged. The main response of the authors was to change what they choose to say about the epidermal ectoderm data. For this part of the manuscript, the message has been changed (mostly) from the claim that a Wnt signal is instructive for planar polarization, to the well justified statement that a Wnt signal can be instructive for planar polarization.*

The new data concern a separate and possibly distinct planar polarization event, so it is not clear how this is supposed to support the findings in the first four figures as opposed to representing a different example. This new data, which was presumably added to address concerns about a true loss of function phenotype, demonstrates that Wnt11b is required for planar polarity of Vangl2 in the neuroectoderm. On the other hand, it does not show that this signal works at a distance, since descendants of the 8 ventral blastomeres into which the MO was injected contribute to the neural plate, and expression of Wnt11b, perhaps at a low level, throughout the neural plate is not ruled out. On the other hand, they do show that ectopic Wnt5a can non-autonomously repolarize neural plate over several cell diameters. They acknowledge that this might occur by either paracrine signaling and propagation, or by graded signaling of the ligand. Concerning this new data, the authors are a bit more circumspect, but shockingly, the final conclusion of the paper again incorrectly claims that Wnt's are instructive for PCP: "Whereas our results demonstrate the instructive role of Wnt proteins in PCP…"

*If we conflate the three different paradigms (ectodermal ectoderm, Wnt11b in neural ectoderm and Wnt5a in neural ectoderm), the data would probably make the best argument yet that a Wnt signal acts instructively to determine PCP, the main weaknesses being that the localized expression of Wnt needs to be more carefully examined, and one would like to see gain and loss of function for the same Wnt (or combination of Wnts) in the same tissue. However, I don't feel it's sufficiently rigorous to combine these systems to come to a conclusion as important as the claim that a Wnt IS instructive for PCP.*

*If the authors wish to try to produce a more definitive conclusion, I'd advise them to build a careful analysis of Wnt11b in the neural plate, and not just add the minimal data presented here as an addendum. If they wish to moderate their overall conclusion to Wnts can instruct planar polarity (two misleading statements would need to be changed, see below), that would be well supported by the data provided. This course would require a somewhat detailed discussion of the distinction between these conclusions. On the other hand, that would be a far less novel result, and one could debate whether it would be of sufficient impact for eLife.*

*The two unjustified statements:*

*"Based on these experiments, Wnt5a appears to act as an instructive cue for Pk3 polarization."*

*This implies that this is what it normally does.*

*"Whereas our results demonstrate the instructive role of Wnt proteins in PCP, whether such function involves mechanical or chemical signaling remains to be clarified in future studies."*

*This is clearly not supported by the present data.*

In the Discussion, we did not mean to imply that Wnt signals are proven to act instructively. Our results only show *how* Wnt proteins behave in our specific experimental conditions. They do not establish that the Wnt-dependent instructive mechanism directs PCP during normal development, but simply suggest a possible scenario of what could happen. The misleading sentences noted by the referee were deleted and corrected, and the above distinction has been made clear in the revised Discussion at the end of the manuscript.

*It's also relevant to comment on one of the responses that the authors provided in the rebuttal.*

*"Several lines of evidence argue for Wnt proteins acting on ectodermal PCP over a long distance during vertebrate gastrulation. First, the effect of ectopic Wnt proteins on the GFP-Pk3/Vangl2 complex is detectable across many cell diameters."*

*So too is clonal knockdown of fz in a fly wing detected several cell diameters away. But this would still clearly be a paracrine signal that is then propagated via other means beyond the clone border. This is therefore not strong evidence for a direct long range action of a ligand.*

*"Second, Wnt5a can reorient endogenous Vangl2, which is normally polarized throughout the neuroectoderm along the anteroposterior axis (new Figure 5)."*

*Doesn't address the paracrine vs. long range issue.*

*"Third, Wnt11b depletion in the posterior region of the embryo disrupts Vangl2 polarity in the neural plate (new Figure 5), supporting its role in long-range anteroposterior PCP."*

*This may be the strongest argument, but still the experimental design is flawed (as noted above), as the vegetal blastomeres of eight cell embryos contribute to neural plate, so the knockdown is presumably distributed throughout the region, and local expression is not ruled out. The authors therefore correctly state "Nevertheless, currently we cannot discriminate between long-range diffusion and local effects of Wnt proteins propagated by a signal relay system or cell division (Alexandre et al., 2014; Farin et al., 2016; Zecca et al., 1996)."*

The experiment has been done with 8-cell embryos in order to deplete Wnt11b as much as possible in its vegetal expression domain (Ku and Melton, 1993). The reviewer correctly pointed out that we do not rule out the effect on local signal production. This is now acknowledged in the revised text (Results and Discussion, eighth paragraph). Overall, our work did not intend to address how far Wnt signals are diffusing. Although our observations are consistent with long-range effects, we make no claims or conclusions regarding this issue in the paper.

Finally, if the authors wish to build a case around Wnt11b. How do they reconcile their story with "Wnt11 has been argued to act permissively in convergent extension during zebrafish gastrulation (Heisenberg et al., 2000)"? Of course, then one is comparing fish and frogs.

The argument that Wnt11 acts permissively in zebrafish embryos was based on the partial rescue of *silberblick* mutants with ectopic Wnt11. We propose that this effect is incomplete because Wnt11 is not properly localized. In other words, the instructive role of Wnt11 may be necessary for the full rescue. This alternative interpretation is fully compatible with our observations. This explanation was introduced into revised Discussion (last paragraph).